# Identification and validation of the common pathogenesis and hub biomarkers in Papillary thyroid carcinoma complicated by rheumatoid arthritis

**Yingming Liu, Xiangjun Kong, Qianshu Sun, Tianxing Cui, Shengnan Xu, Chao Ding** *

General Surgery Ward four, Department of General Surgery, The Second Affiliated Hospital of Harbin Medical University, Harbin, China

* dingchao851221@163.com (CD)

## Abstract

### Background

Papillary thyroid carcinoma coexisting with rheumatoid arthritis is frequently observed in clinical patients, yet its pathogenesis has not been fully elucidated. This investigation sought to further explore the molecular underpinnings of these two diseases.

### Methods

Gene expression profiles for thyroid papillary carcinoma and rheumatoid arthritis patients were obtained from the Comprehensive Gene Expression Database (GEO). Following the discovery of shared differentially expressed genes (DEGs) between these two conditions, three separate analyses were conducted. These included functional annotation, the establishment of a protein–protein interaction (PPI) network and module, and the identification of hub genes via coexpression analysis. The final step involved the validation of target genes via clinical specimens.

### Results

This study analyzed datasets from four GEO databases and identified 64 common DEGs. Functional enrichment analysis revealed that these genes are predominantly associated with pathways related to immunity and signal transduction. Protein–protein interaction (PPI) network analysis revealed complex interactions among these differentially expressed genes and highlighted several genes that may play pivotal roles in shared pathological mechanisms, namely, CCR5, CD4, IL6, CXCL13, FOXM1, CXCL9, and CXCL10.

### Conclusion

Our study highlights the shared pathogenesis between papillary thyroid cancer and rheumatoid arthritis. Shared pathways and crucial genes could offer novel perspectives for subsequent investigations into the mechanisms of these diseases.

**Data availability statement:** All relevant data are within the paper and its Supporting Information files.

**Funding:** The author(s) received no specific funding for this work.

**Competing interests:** The authors have declared that no competing interests exist.

## Introduction

Papillary thyroid carcinoma (PTC), which represents the majority of differentiated thyroid cancers, accounts for more than 90% of all thyroid cancer cases in adults [1]. Although PTC is generally known as a slow-progressing tumor, some of these cancer cells can spread to the lymph nodes near the thyroid gland. This process mainly involves central lymph node metastasis (LNM) and lateral cervical LNM [2]. The occurrence of lateral cervical lymph node metastasis often leads to a worse prognosis and necessitates a change in the treatment regimen [3]. As such, accurate preoperative evaluation of lateral cervical lymph node metastasis holds significant clinical importance.

Rheumatoid arthritis (RA), a prevalent chronic inflammatory disease, predominantly affects joints [4]. RA is more accurately characterized as a syndrome with extra-articular manifestations, including rheumatoid nodules, lung involvement, vasculitis, and systemic comorbidities. The prevalence of RA varies from 0.5% to 1%, with a discernible decline from north to south and from urban to rural regions [5]. The previous decade has seen a significant shift in RA treatment. This transformation, marked by the introduction of novel treatments, the commencement of early intervention, the development of updated classification standards, and the application of efficient therapeutic strategies, has significantly improved outcomes both at the joint level and systemically.

Patients with both rheumatoid arthritis and papillary thyroid cancer are not uncommon in the clinical setting (S1 Table). A study revealed that thyroid cancer is associated with eight different autoimmune diseases [6]. Although the co-occurrence of these two diseases is relatively frequent in clinical settings, the precise mechanism of their coexistence remains unclear. Originally, interferon-γ-induced protein 10 (CXCL10) was characterized as a chemokine stimulated by IFN-γ and released by a variety of cell types [7]. CXCL10, which is functionally categorized as an inflammatory chemokine, can inhibit angiogenesis and serve as a vascular inhibitory chemokine when it lacks the ELR motif [8]. CXCL10 binds to CXCR3, orchestrating immune responses by activating and recruiting leukocytes [9,10]. However, CXCR3 expression is not limited to immune cells, and it is also found in resident cells, perivascular cells, and mesangial cells. The recognized ligands that bind to CXCR3 include CXCL9, CXCL10, and CXCL11. Recent studies suggest an increase in CXCL10 expression levels in both serum and tissue across a range of autoimmune diseases. These include systemic sclerosis (SSc), rheumatoid arthritis (RA), and multiple sclerosis (MS) [11–15]. These observations suggest potential pathophysiological pathways common to these diseases.

Interleukin 6 (IL-6), a cytokine with multiple and overlapping functional activities, is a representative member of the cytokine family. All these cytokines utilize the common IL-6 signal transducer gp130 [16]. IL-6 production can be triggered by infection and various forms of inflammation. Specifically, IL-6 is predominantly and swiftly generated by macrophages in response to pathogens or inflammation-associated damage-related molecular patterns. IL-6 plays a protective role by initiating acute phase and immune responses to eliminate infectious agents and repair damaged tissues [17]. In addition, IL-6 plays a vital role in both innate and adaptive immunity. It is produced by several cell types, with notably increased production at inflammation sites [18]. In the event of infection, toll-like receptors identify bacteria and viruses, which results in the direct or indirect stimulation of IL-6 and other inflammatory cytokines. Importantly, the synthesis of IL-1 and TNFα also initiates the production of IL-6 [19]. In summary, the dysregulation of IL-6 production results in chronic inflammation. Additional immune pathways also contribute to the progression of RA, ultimately causing the overexpression of IL-6. For example, the NF-kB pathway in activated B cells plays a significant role as an inflammatory mediator in RA, resulting in an increase in TNFα, which in turn increases IL-6 levels. IL-6 might also subtly contribute to the development of RA; observations

have shown that IL-6 is involved in cytokine release syndrome associated with T-cell therapy. In such instances, blocking IL-6 has resulted in positive outcomes, underscoring the central role of IL-6 in inflammatory syndromes [20,21]. IL-6 seems to be linked to other systemic symptoms related to RA. In patients with RA, the levels of IL-6 in the serum and synovial fluid are generally increased in the impacted joints [22]. Moreover, research has revealed that IL-6 can increase the proliferative capacity of PTC cells. In addition, a negative relationship exists between IL-6 and the expression of sodium/iodide symporters in thyroid cancer tissues. The uptake of iodine mediated by NIS is essential for the effectiveness of radioactive iodine therapy [23]. These findings imply that IL-6 could play a significant role in the onset and progression of PTC.

The transcriptional characteristics between PTC and RA may provide new insights into the common pathogenesis of these two diseases. The main goal of this study to identify the key genes that play a significant role in the pathogenesis of PTC when it with RA. To accomplish this, we conducted a detailed analysis of four gene expression datasets downloaded from the Gene Expression Omnibus database. We used comprehensive bioinformatics and enrichment analysis to identify the common differentially expressed genes (DEGs) and to understand their functions. Moreover, we established a PPI network for the examination of gene modules and pinpointing essential genes. At the end of our analysis, we identified 16 key genes. We then analyzed the transcription factors of these genes and verified their expression. Importantly, we conducted clinical sample verification of CXCL10, confirming its crucial role in the onset and progression of both diseases. The key genes between PTC and RA identified in this study are expected to provide new insights into the biological mechanisms underlying these two diseases.

## Materials and methods

### Data source

The Gene Expression Omnibus (http://www.ncbi.nlm.nih.gov/geo) is a publicly accessible database that houses a vast array of high-throughput sequencing and microarray datasets. To find related gene expression datasets, we used 'papillary thyroid cancer' and 'rheumatoid' as our search keywords. We established the following criteria for inclusion: the datasets should originate from two separate expression profiles that were produced on identical sequencing platforms and encompass the maximum sample size. Moreover, the samples tested within these datasets should be obtained from human subjects. As a result of our search, we downloaded four microarray datasets: GSE165724 [24], GSE64912 [25], GSE55457 [26], and GSE55235 [26]. The datasets were created with the use of the Affymetrix GPL570 platform, alternatively known as the Affymetrix Human Genome U133 Plus 2.0 Array. The GSE165724 dataset includes 28 paired samples from papillary thyroid cancer patients, with 12 normal samples and 16 PTC samples. The GSE64912 dataset contains the same number of samples as the GSE165724 dataset. The GSE55475 dataset is comprised 10 normal samples and 13 RA samples. Finally, the GSE55235 dataset consists of samples from synovial tissue, including 10 normal samples and 10 RA samples.

### Identification of DEGs

R software (version 4.1.3; accessible at https://www.r-project.org/), in conjunction with the Bioconductor software package (available at http://www.bioconductor.org/), was used for the correction and analysis of the original data. The RNA-seq data were processed and normalized via the DESeq2 package. The repeatability of the GSE102485 data was verified through the application of principal component analysis (PCA). The standard for statistical significance is

defined as a |log2FC| value exceeding 1 and an adjusted P value falling below 0.05. The differentially expressed genes (DEGs) are represented in a volcano plot created using the "ggplot2" software package.

## Enrichment analyses of DEGs

Gene Ontology (GO) is a wide-ranging database that delivers unambiguous annotations of gene products, including their functions, the biological pathways they engage in, and their intracellular locations. The Kyoto Encyclopedia of Genes and Genomes (KEGG) pathway is a specific database. The KEGG Orthology-Based Annotation System (KOBAS) functions as a web server built for the functional annotation and enrichment of genes/proteins. Functional annotation data from 4325 species were collected. The results of the GO and KEGG pathway enrichment analyses were sourced from the KOBAS 3.0 database. An adjusted P value less than 0.05 was considered statistically significant. Together these resources establish a sturdy framework for the exploration and understanding of complex biological systems and processes.

## PPI network construction and module analysis

The Search Tool for the Retrieval of Interacting Genes (STRING; http://string-db.org) (version 11.0) is an effective resource capable of exploring the relationships between proteins of interest. This includes direct binding relationships or the coexistence of upstream and downstream regulatory pathways. Interactions with a combined score greater than 0.4 are recognized as statistically significant. The PPI network was constructed via Cytoscape (http://www.cytoscape.org) (version 3.7.2). Molecular complex detection (MCODE) technology is utilized to analyze crucial functional modules. The selection criteria are defined as follows: K-core = 2, degree cutoff = 2, max depth = 100, and node score cutoff = 0.2. The KEGG and GO analyses of the genes present in these modules were subsequently performed via KOBAS 3.0.

## Selection and analysis of hub genes

Hub genes were identified via the cytoHubba plug-in of Cytoscape. In this operation, seven frequently used algorithms (MCC, MNC, Degree, Closeness, Radiality, Stress, EPC) are applied to evaluate and select the hub genes. GeneMANIA (http://www.genemania.org/), a robust tool adept at uncovering internal connections within gene sets, was subsequently used to construct a coexpression network of these identified hub genes.

## Validation of hub gene expression in other datasets

The mRNA expression levels of the identified hub genes were cross-verified via two distinct datasets: GSE33630 [27,28] and GSE77298 [29]. The GSE33630 dataset is comprised of 45 samples from individuals in good health and 49 samples from patients suffering from PTC. The GSE77298 dataset is comprised of 7 samples from healthy individuals and 16 samples from patients with RA. An analysis comparing the two datasets was performed via the t test. A P value less than 0.05 was used as the benchmark for statistical significance. This rigorous verification procedure ensures the reliability of the hub genes identified in the context of these diseases.

## Prediction and verification of transcription factors (TFs)

Transcriptional regulatory relationships revealed by sentence-based text mining (TRRUST) acts as a comprehensive resource for predicting transcriptional regulatory networks. TRRUST provides information on target genes linked to transcription factors (TFs) and the regulatory relationships between these TFs. At this time, TRRUST incorporates data from two species:

humans and mice. TRRUST includes 8,444 and 6,552 TF-target regulatory relationships for 800 human TFs and 828 mouse TFs, respectively. Transcription factors that regulate the pinpointed hub genes were obtained from the TRRUST database. An adjusted P value less than 0.05 was recognized as statistically significant. The expression levels of these transcription factors were subsequently validated in two datasets, GSE165724 and GSE55235, via a t test.

## Immunofluorescence staining and confocal microscopy

Immunofluorescence staining was performed on samples from both the PTC-RA (n = 3) and PTC (n = 3) groups using anti-CXCL10 antibodies. This study was approved by the Medical Ethics Committee of the Second Affiliated Hospital of Harbin Medical University, and the ethics approval number is KY2024-002. All participants in the study provided informed consent. Each section was independently assessed by two researchers via an LSM 800 confocal microscope (Carl Zeiss MicroImaging GmbH, Jena, Germany).

## Quantitative real-time polymerase chain reaction (QPCR)

Total RNA was extracted from the samples using a commercial RNA extraction kit (RNeasy Mini Kit, Qiagen). cDNA was synthesized from 1 μg of total RNA using a reverse transcription kit (High-Capacity cDNA Reverse Transcription Kit, Thermo Fisher Scientific). QPCR was conducted using a real-time PCR system (ABI 7500 Fast Real-Time PCR System, Roche) with SYBR Green PCR Master Mix (SYBR Green Master Mix, Thermo Fisher Scientific) for DNA amplification and detection. The primers for CXCL10 and GAPDH were as follows: CXCL10 Forward primer: 5′-GTGGATGTTCTGACCCTGCT-3′, CXCL10 Reverse primer: 5′-GGAGGATGGCAGTGGAAGTC-3′, GAPDH Forward primer: 5′-AATGGGCAGCCGTTAGGAAA-3′, GAPDH Reverse primer: 5′-GCGCCCAATACGACCAAATC-3′. The relative expression levels of CXCL10 were normalized to the endogenous control GAPDH using the normalization method. Each sample was run in triplicate to ensure technical reproducibility, and the mean Ct values were used for further analysis.

## Western blot analysis

Protein extracts were prepared using a standard lysis buffer containing protease inhibitors. Equal amounts of protein were separated by sodium dodecyl sulfate-polyacrylamide gel electrophoresis (SDS-PAGE) and transferred onto a polyvinylidene fluoride (PVDF) membrane. Membranes were blocked with 5% non-fat dry milk in Tris-buffered saline with Tween 20 (TBST) for 1 hour at room temperature and then incubated with primary antibodies against CXCL10 (Abcam, ab124817, dilution 1:1000) and GAPDH (Sigma-Aldrich, G9545, dilution 1:2000) overnight at 4°C. After washing with TBST, membranes were incubated with horseradish peroxidase-conjugated secondary antibodies (dilution 1:5000) for 1 hour at room temperature. Immunoreactive bands were visualized using a chemiluminescent substrate (Thermo Fisher Scientific) and detected with a gel documentation system. Densitometric analysis of the bands was performed using imagJ, and the relative expression of CXCL10 was normalized to GAPDH.

## Statistical analysis

The data were evaluated via GraphPad Prism software and are presented as the means ± standard deviations (SDs). Differences between two groups were assessed via an unpaired t test with Welch's correction. When data from more than two groups were compared, analysis of variance

(ANOVA) was utilized. A P value of less than 0.05 was recognized as statistically significant. This rigorous statistical analysis ensures the reliability of the results obtained in this study.

## Results

### Identification of differentially expressed genes

The research flowchart of this study is shown in Fig 1. Our comprehensive study preprocessed four distinct datasets, namely, GSE55235, GSE55457, GSE165724, and GSE64912. These datasets included both normal and rheumatoid arthritis (RA) samples derived from synovial tissue, as well as normal and papillary thyroid carcinoma (PTC) samples obtained from thyroid tissue. To ensure the utmost reliability and quality of our data, we employed two distinct yet complementary methods. The first method involved an examination of the relative expression levels of housekeeping genes, such as GAPDH and ACTB, with those of ordinary genes. Our findings from this analysis indicated that the expression levels of housekeeping genes were significantly elevated compared with those of regular genes, suggesting their crucial role in maintaining cellular functions. The second method involved assessing whether a substantial difference in the measured gene expression levels existed between the samples. Our rigorous analysis revealed that the gene expression levels across the samples were essentially consistent, indicating that our data are of high quality and can be relied upon for further investigation.

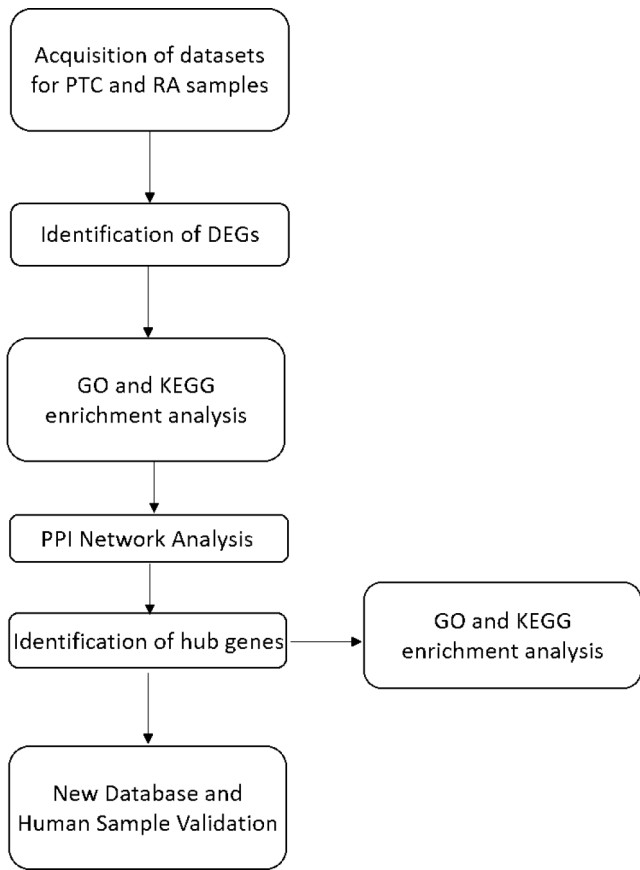

**Fig 1. Research design flowchart.**

Following the preprocessing stage, we performed differential analysis of gene expression differences in both RA and PTC samples. The results of this analysis revealed that after filtering out genes with low expression, we tested a total of 1422 genes in the GSE55235 dataset for RA samples. Among these genes, 819 presented increased expression in RA samples, whereas 691 presented decreased expression. In the GSE55457 dataset, we tested a total of 14,826 genes, with 477 genes found to be upregulated and 391 genes downregulated in RA samples. For the PTC samples in the GSE165724 dataset, we tested a total of 21328 genes. Among these genes, 2716 genes were upregulated in the PTC samples, whereas 2326 genes were downregulated. In the GSE64912 dataset, we tested a total of 15397 genes, with 1304 genes found to be upregulated and 526 genes downregulated in PTC samples (Fig 2A and 2B).

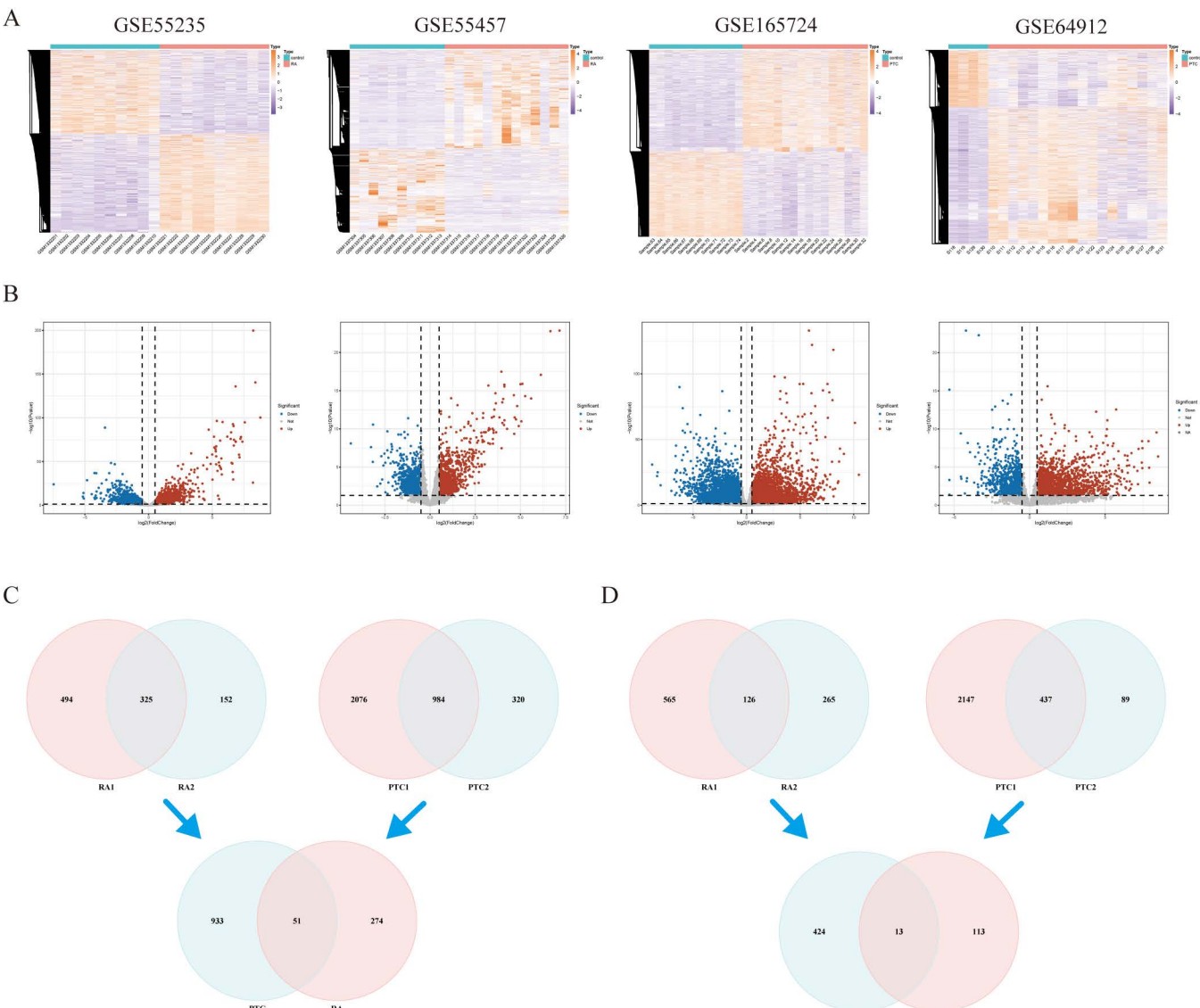

**Fig 2. Identification of DEGs in PTC and RA.** (A) Heatmap of A) Heatmap of differential gene expression. (B) Volcano map. The upregulated genes are marked in red; the downregulated genes are marked in blue. (C-D) The four datasets showed an overlap of 51 upregulated DEGs (C) and 13 downregulated DEGs (D).

In an effort to gain deeper insights, we further analyzed the genes commonly altered in the two diseases across the four databases. Our findings revealed that there were 64 commonly differentially expressed genes: 51 genes presented increased expression, whereas 13 genes presented decreased expression in both diseases (as shown in Fig 2C and 2D). These commonly differentially expressed genes could play crucial roles in the onset and progression of both diseases.

## Functional enrichment analysis

To gain a deeper understanding of the biological functions influenced by the 64 differentially expressed genes that are commonly altered in rheumatoid arthritis (RA) and papillary thyroid carcinoma (PTC), we conducted a comprehensive functional enrichment analysis. For the upregulated genes, we performed a Gene Ontology (GO) analysis. GO analysis revealed the enrichment of a total of 219 pathways, revealing a broad spectrum of potential biological functions influenced by these genes (Fig 3A, 3B and 3E). These enriched pathways were associated primarily with immune-related pathways and signal transduction pathways, suggesting a potential link between these genes and the immune response or signal transduction mechanisms. For the downregulated genes, we conducted a GO analysis, which revealed the enrichment of 5 pathways, predominantly related to endocrine- or metabolic-related pathways (Fig 3C and D). These results provide invaluable insights, aiding in understanding the impact of these differentially expressed genes on various biological functions.

## Protein-protein interaction (PPI) network analysis

To explore the intricate relationships between the 64 differentially expressed genes, we utilized the comprehensive STRING database. Following a meticulous integration and filtering process, we retained 55 genes in the final PPI network (Fig 4A and 4B). In this network, the upregulated genes are represented in blue, symbolizing their increased activity, whereas the downregulated genes are depicted in red, indicating their decreased activity. We employed the MCODE tool in Cytoscape for module analysis, with the aim of identifying tightly interconnected regions within the network. This analysis yielded a total of 3 modules, each representing a potential functional unit within the network (Fig 4C). We subsequently used the cytoHubba tool in Cytoscape for network analysis. By ranking the genes on the basis of their degree, we pinpointed the five most central Hub genes in the network. These key players are CCR5, CD4, IL6, CXCL13, FOXM1, CXCL9, and CXCL10. Owing to their high connectivity, these genes may play pivotal roles in the shared pathological mechanisms of RA and PTC. Finally, we performed an association analysis on these hub genes via GeneMANIA. The results revealed a close relationship between these hub genes (Fig 4D), suggesting their potential synergistic roles in the biological processes being studied. This implies that these genes may not function in isolation but rather work together in a coordinated manner to influence disease pathogenesis. This comprehensive PPI network analysis not only elucidates the complex interactions among the differentially expressed genes but also highlights the key genes that may play pivotal roles in the shared pathological mechanisms of RA and PTC.

## Hub gene functional enrichment analysis

To delve deeper into the functional roles of the identified hub genes, we performed a comprehensive functional enrichment analysis. In the GO analysis, we successfully enriched a total of 365 pathways (Fig 4E), revealing a broad spectrum of potential biological functions influenced by these hub genes. On the other hand, in the KEGG analysis, we enriched a total of 19 pathways (Fig 4F), offering a more focused perspective on the potential metabolic pathways

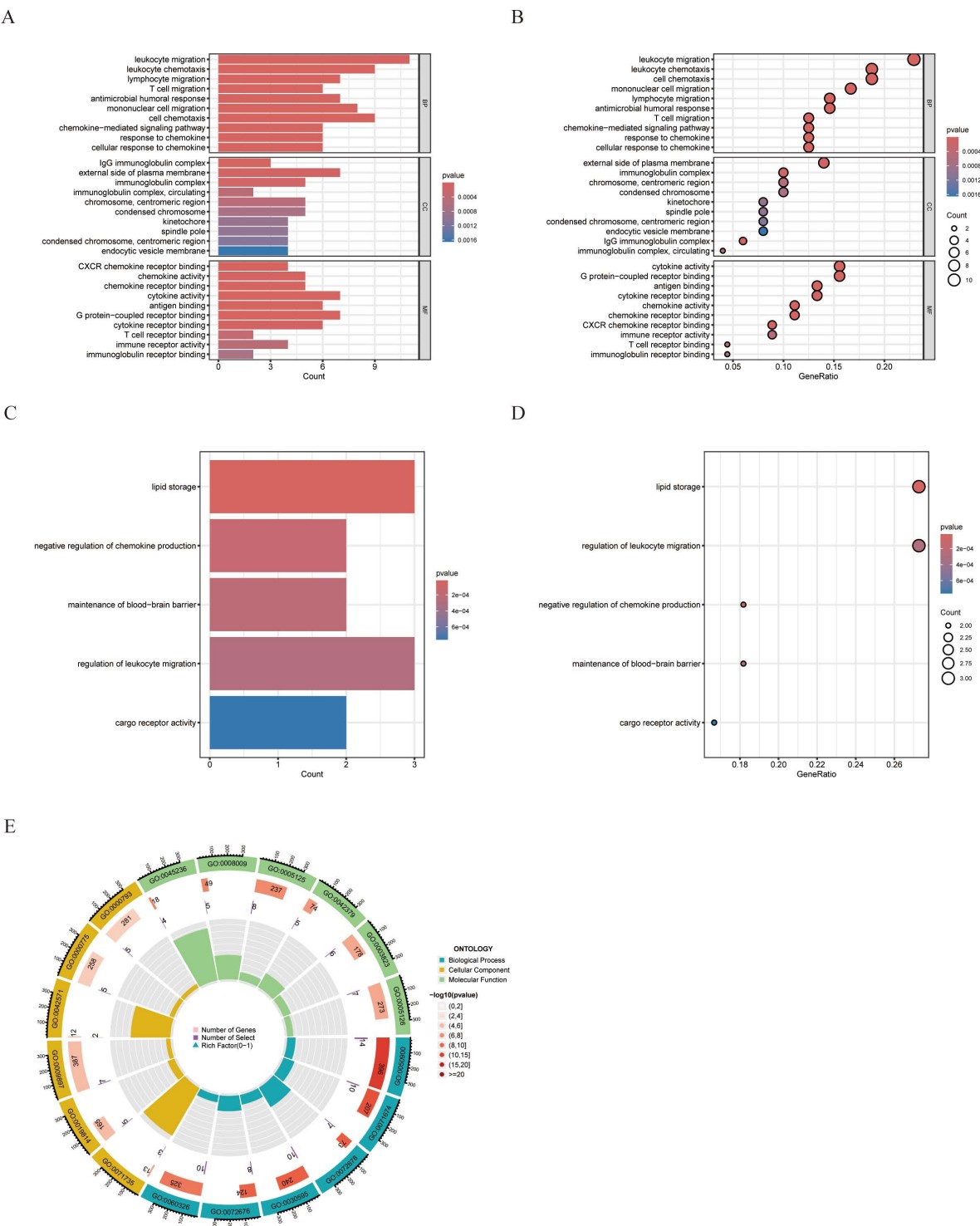

**Fig 3. Common DEGs functional enrichment analysis results.** (A-B) (A-B) The enrichment analysis results of GO pathway for upregulated genes. (C-D) The enrichment analysis results of GO pathway for downregulated genes. (E) Circle diagram of GO pathway analysis for upregulated genes.

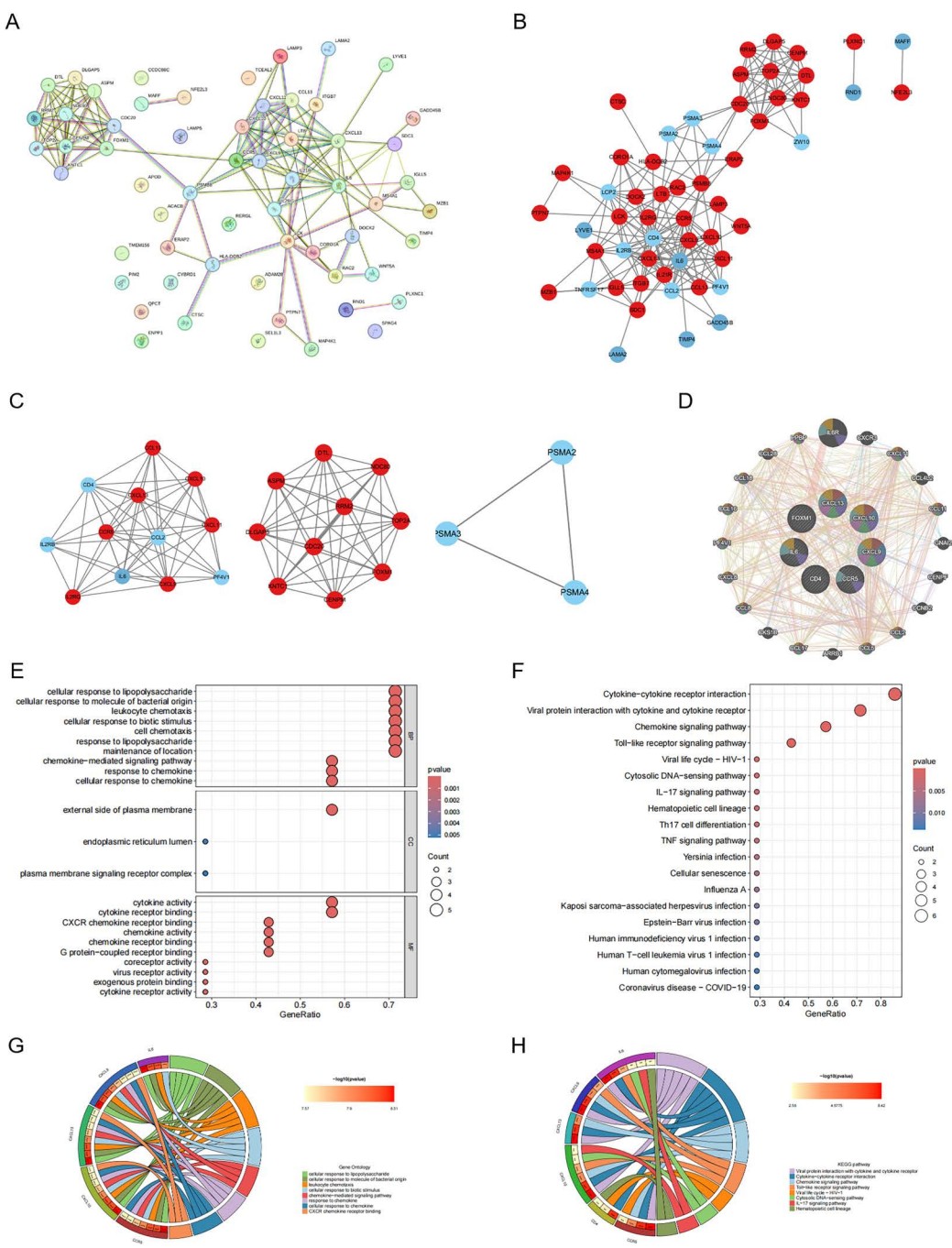

**Fig 4. PPI and Co-expression network construction and significant gene module of the modular genes and functional analysis of hub genes.** (A, B) PPI Network Diagram. Green indicates upregulated genes, and red indicates downregulated genes. (A) Before filtering out isolated genes. (B) After filtering out isolated genes. (C) Three significant gene clustering modules. (D) Hub genes and their coexpressed genes. (E–H) GO and KEGG enrichment analyses of the hub genes. (G, H) The outermost circle represents the term on the right, and the inner circle on the left represents the significant p value of the corresponding pathway of the gene.

influenced by these genes. Interestingly, a significant proportion of these enriched pathways were related to immune or signaling pathways. In addition to the functional enrichment analyses, we also performed an association analysis on the hub genes. The results, visualized in a circle plot, revealed a close relationship between these hub genes (Fig 4G and 4H), suggesting their potential synergistic roles in the biological processes under study. Notably, the immune-related gene CXCL10 was found to be involved in these pathways, indicating its potential role in the underlying biological processes.

This comprehensive functional enrichment and association analysis provides valuable insights into the potential roles of these hub genes. The analysis results revealed a crucial gene, CXCL10, which is anticipated to have a significant effect on disease progression.

## Validation of analysis results using a new dataset

In an effort to corroborate our initial analysis results, we sought out a new dataset and embarked on a validation process to examine the expression of hub genes. For this rigorous validation process, we selected two distinct datasets, namely, GSE77298 for rheumatoid arthritis (RA) and GSE33630 for papillary thyroid carcinoma (PTC). Upon analyzing these validation sets, we discovered that only three genes (CD4, CXCL13, and IL6) were detected as hub genes, indicating their potential significance in disease processes. We then proceeded to examine the expression of these three genes in the two diseases. Our findings are as follows: the p value of CXCL13 in the PTC validation set did not reach a significant level (P = 0.22). However, the remaining genes, CD4 and IL6, exhibited significant differences in expression (P < 0.05) (Fig 5). Furthermore, these genes were more highly expressed in the disease group than in the normal samples, suggesting their potential role in disease pathogenesis. These results support our previous findings, affirming that these hub genes exhibit significant expression differences in the two diseases. This validation procedure emphasizes the solidity of our preliminary analysis and highlights the potential influence of these hub genes on disease development.

## Investigation of hub gene regulators

In our continued exploration of the hub genes, we utilized the comprehensive TRRUST database to identify the six most pertinent regulators of the hub genes. These regulators include CREB1, MYC, RELA, NFKB1, and SP1. Subsequently, we embarked on a validation process to examine the expression of these regulators within the original dataset, comparing the disease group and the healthy group. Our findings revealed that these regulators generally exhibited significant expression differences between the normal and disease groups. Interestingly, all of these genes presented increased expression in the disease group, suggesting their potential role in disease pathogenesis. Upon further analysis, we found that MYC and STAT1 were the most relevant regulators, corresponding to FOXM1 and IL6 and to IL6 and CXCL10, respectively. Additionally, RELA and SP1 were related to CXCL10, IL6, CCR5, and FOXM1. These findings imply that these regulators could influence the regulation of these hub genes. Notably, we discovered that among all the hub genes, IL6 and CXCL10 were the most relevant to the changes in regulators. These findings suggest that IL6 and CXCL10 could be crucial in triggering and promoting these two conditions.

## Validation of CXCL10 expression

In the course of our comprehensive functional enrichment analysis, we discovered that CXCL10, a gene that is closely associated to immune responses, is prominently featured in the pathways that are most enriched. To substantiate these intriguing findings, we comprehensively validated CXCL10 expression in both PTC-RA and PTC samples. During this rigorous

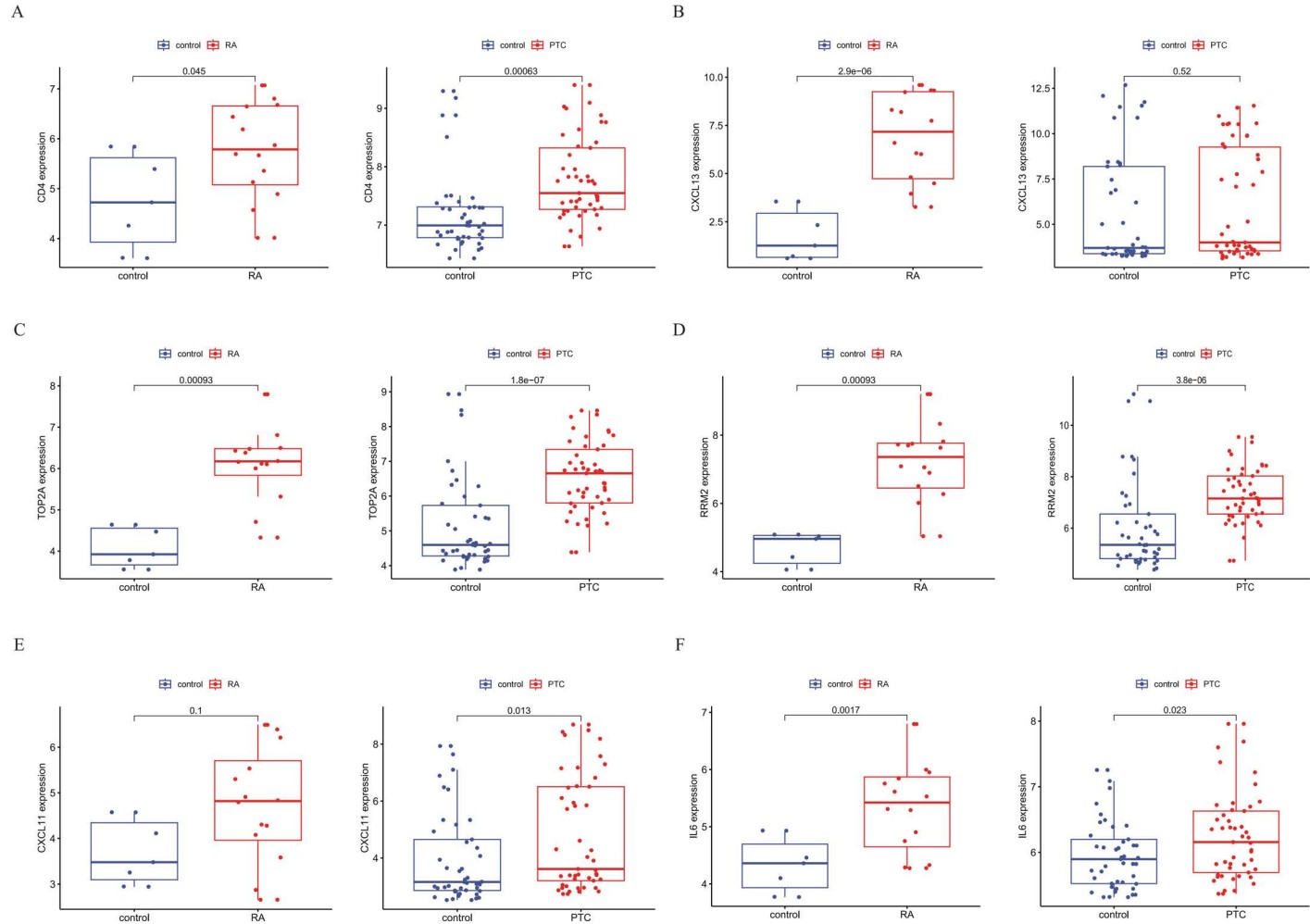

**Fig 5. The relative expression level of hub genes.** (A), CXCL13 (B), TOP2A (C), RRM2 (D), CXCL11 (E), and IL-6 (F) between the Two Disease Groups and the Normal Group.

validation process, an anti-CXCL10 antibody was utilized for immunofluorescent staining, a sophisticated technique that enables the visual observation of CXCL10 expression in the samples. The staining results, which revealed a noticeable increase in the expression level of CXCL10 in the diseased tissues (Fig 6 A–6D), were highly consistent with our initial analysis. Simultaneously, we conducted QPCR validation, and the results were similar, with the expression level of CXCL10 in the PTC+RA group being significantly higher than that in the PTC group. The potential importance of CXCL10 in these diseases and the robustness of our findings are emphasized by this consistency. These results further supported our discovery that the immune-related gene CXCL10 is significantly differentially expressed in both diseases. These findings could pave the way for new insights into the shared pathological mechanisms of these two diseases.

## Discussion

This study delves into the pathogenic association between PTC and RA. The simultaneous occurrence of PTC and RA was explained by the processes we discovered, and nine key hub

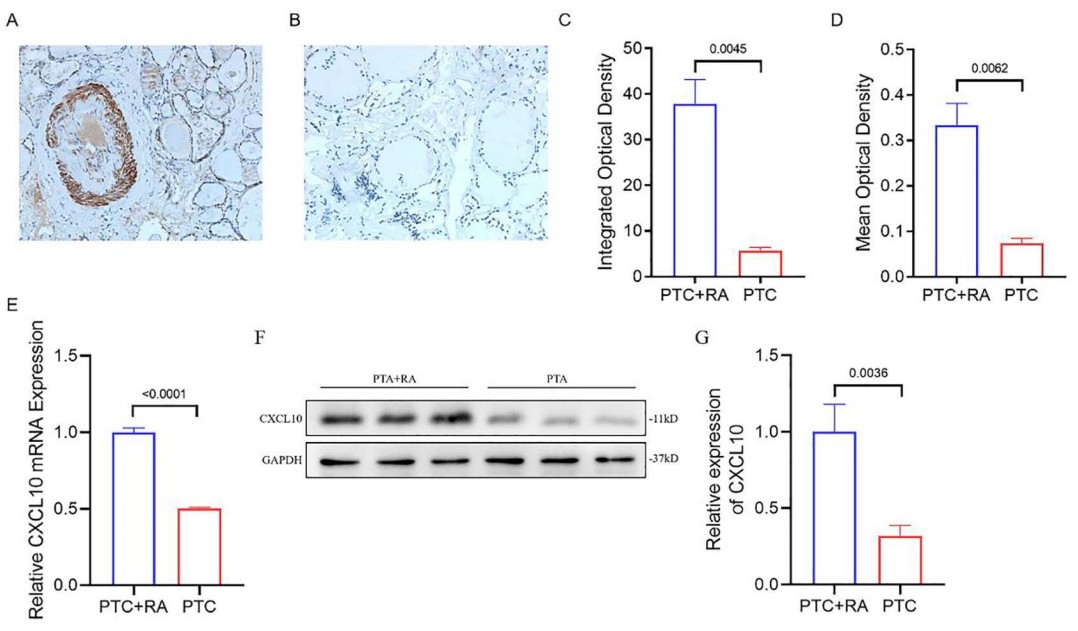

**Fig 6. The expression of CXCL10 in tissues.** (A, (A, B) Immunohistochemical staining of CXCL10 in PTC+RA samples (A) and PTC samples (B). (C, D) The integrated optical density (C) and mean optical density (D) of CXCL10 staining in PTC+RA samples and PTC samples. (E) QPCR validation of CXCL10 relative expression.

genes were identified. An increasing body of research has reported that patients suffering from rheumatic diseases often exhibit thyroid disease, with a particularly close relationship observed between rheumatoid arthritis and thyroid disease [30]. A study from China demonstrated a strong association between rheumatoid arthritis and thyroid cancer [31]. However, the exact cause of the link between PTC and RA remains elusive. Theoretically, an underlying imbalance in the immune system could lead to the progression of cancer [32]. Therefore, RA, a disease of the immune system, is speculated to be associated with PTC. This research aimed to elucidate the basic pathogenic associations and mechanisms between PTC and RA, ultimately enriching our knowledge of these diseases.

This research provides a fresh view of the potential overlapping pathogenic pathways between PTC and RA. Through GO enrichment analysis of the identified hub genes, significant enrichment in endocrine metabolism and the immune response was observed. Furthermore, the KEGG enrichment analysis of the shared DEGs revealed significant enrichment in a few crucial signaling pathways: the PPAR signaling pathway, the AMPK signaling pathway, and the adipocytokine signaling pathway. The PPAR signaling pathway, which is mainly involved in the energy metabolism pathway, plays a crucial role in preserving both the cellular and systemic energy balance [33] as well as in modulating innate or acquired adaptive immunity and immune-mediated inflammation [34]. The AMPK signaling pathway, which is critical for maintaining energy balance and metabolism, including aspects such as caloric restriction, aging, and obesity [35], is the target of numerous small molecule drugs, such as metformin [36]. This pathway plays a dual role acting as the guardian of the cellular energy balance and functioning as an important participant in the role of organelles in the body [35]. These findings underscore the complex interplay of metabolic and immune pathways in the pathogenesis of PTC and RA and could pave the path for the development of novel therapeutic approaches aimed at these diseases. Historically, the AMPK protein was viewed as a tumor

inhibitor because of its ability to inhibit the mTORC1 signaling pathway through the phosphorylation of TSC2 and RAPTOR, which are pathways that are often associated with carcinogenesis [37]. Specifically, AMPK has been shown to phosphorylate and destabilize GLI1, thereby inhibiting its carcinogenic activity in medulloblastoma [38]. However, recent studies have revealed the potential carcinogenic activity of AMPK, suggesting that it may promote cancer cell survival and the progression of breast cancer [39]. The adipokine signaling pathway, a crucial metabolic pathway within the body, can lead to many diseases, such as insulin resistance, glucose intolerance, hypertension, and dyslipidemia, when it becomes dysregulated [40]. Moreover, research has indicated that the adipokine signaling pathway may play a role in the onset of thyroid diseases [41]. In this study, we scrutinized the mRNA expression patterns of key genes in human RA datasets and PTC samples. The analysis revealed notable upregulation of two central genes, CXCL10 and IL6, in both the PTC and RA cohorts. These data imply a potential overlap in the pathways involved in the onset and progression of PTC when complicated with RA, offering fresh insights into the interconnected etiology of these two conditions. Among the group of key genes, CXCL10 is produced by both immune and nonimmune cells in response to inflammatory triggers [35]. CXCL10, a protein that is functionally classified as a Th1-chemokine, weighs 10 kDa and interacts with the CXCR3 receptor and modulates immune reactions by activating and recruiting leukocytes [7]. In RA patients, CXCL10 has been identified in the serum and synovial tissue (ST) [11]. CXCL10 also acts as a powerful immune enhancer in RA. The elevated presence of CXCL10 in RA synovial fluid stimulates the generation of RANKL, thereby increasing osteoclast formation [42]. CXCL10 has also been linked to cancer. Some studies have demonstrated that CXCL10 contributes to the generation of a "hot" tumor [43], supporting tumor development by drawing tumor cells to establish metastasis [44]. Shi and colleagues verified that CXCL10 expression can serve as an indicator for anti-PD-1/PD-L1 treatment [45]. CXCL10 can also be used as a potential target for certain chemotherapeutic agents [46]. This study not only demonstrated that the chemokine CXCL10 can serve as a prognostic marker for pancreatic cancer but also shed light on the role of the tumor microenvironment [47]. IL-6, a crucial cytokine with immunomodulatory properties, has been shown to impact the development of numerous diseases. These conditions include autoimmune disorders, chronic inflammation, and even malignancies [48]. The expanding comprehension of IL-6 biology has resulted in the identification of three unique forms of IL-6-driven signaling: traditional IL-6 signaling, IL-6 trans-signaling, and IL-6 cluster signaling. In traditional signaling, IL-6 attaches to mIL-6R to stimulate a signal-transducing homodimer of the gp130 receptor chain [49]. This differs from IL-6 trans-signaling, where soluble variants of IL-6R, produced through either alternative splicing or restricted proteolytic processing, create complexes with IL-6 to stimulate membrane-anchored gp130 [50]. Finally, in IL-6 cluster signaling, IL-6–mIL-6R complexes established on a transmitting cell activate gp130 subunits on an adjacent receiving cell [51].

Our comprehensive functional enrichment analysis revealed that CXCL10, a gene closely related to immune responses, was significantly enriched in the most highly enriched pathways. This finding strongly implies that CXCL10 could have a significant influence on the development and progression of illnesses, indicating its potential importance in the fields of disease research and therapeutic development. Given the important role of CXCL10, we chose CXCL10 for subsequent experimental verification. Pathological sections from two groups of patients were selected: one group with both rheumatoid arthritis and papillary thyroid cancer and the other group with only papillary thyroid cancer. The expression level of CXCL10 was verified via immunohistochemistry. At the same time, we also conducted PCR and WB experiments to further confirm the expression level of CXCL10. The data from the experiments indicated that CXCL10 was highly expressed in both cohorts, with a statistically meaningful

elevated expression level in the mixed disease group. These findings suggest that CXCL10 plays a significant role in the occurrence and development of both thyroid papillary carcinoma and rheumatoid arthritis. This comprehension could conceivably lay the groundwork for novel treatment approaches aimed at these conditions. In addition, these findings further highlight the necessity of additional investigations into the function of CXCL10 in disease development and therapy.

Given the strong correlation between thyroid disease and rheumatoid arthritis, this study takes a unique approach compared with previous studies. This study places greater emphasis on the exploration of hub genes and related transcription factors. A complex network of interactions was constructed to identify key nodes associated with DEGs. This bioinformatics methodology has demonstrated its dependability across a broad range of illnesses [52]. In addition, a study of pertinent transcription factors was carried out, and their expression levels were confirmed in the initial dataset. The expression level of CXCL10 was corroborated in clinical samples and was found to be considerably elevated in patients with both conditions, more so compared with patients with only papillary carcinoma but less so compared with patients with multiple diseases. These insights are anticipated to offer potential pathways for future investigations into the molecular mechanisms of papillary thyroid carcinoma in conjunction with rheumatoid arthritis. In addition, these findings could lay the groundwork for the development of novel therapeutic approaches aimed at treating these diseases.

In this study, we investigated the shared pathogenesis between PTC and RA and identified key genes and their interconnected roles. Our findings offer novel insights and perspectives compared with existing research within this domain. First, our identification of 64 DEGs that are common to both PTC and RA, along with our functional enrichment and PPI network analyses, provides a deeper understanding of their roles in the immune response and signal transduction pathways. Notably, our emphasis on the potential synergistic roles of CXCL10 and IL6 in both diseases is a novel contribution that has not been extensively explored in previous studies. Second, our comprehensive bioinformatics approach, coupled with the validation of our findings through clinical specimens, strengthens the reliability of our results. This stands in contrast to some studies that rely on a single data source or lack clinical validation, offering a more robust and thorough investigation. Furthermore, our exploration of potential shared pathological pathways, such as the PPAR signaling pathway, AMPK signaling pathway, and adipocytokine signaling pathway, highlights the intricate interplay between metabolic and immune pathways in the pathogenesis of PTC and RA. This emphasis on the crosstalk between these pathways provides a new perspective for understanding the complex mechanisms underlying these diseases and could pave the way for the development of novel therapeutic strategies targeting these pathways. Finally, the validation of our findings in independent datasets further substantiates the significance of the identified hub genes in PTC and RA. This finding reinforces the generalizability and clinical relevance of our conclusions, addressing concerns about the reproducibility of results that some studies may face. In summary, by revealing the common pathological pathways and key genes between PTC and RA, our study not only contributes new perspectives to the field but also provides potential biomarkers for future diagnostics and therapeutics. We anticipate that these findings will stimulate further research into the interconnected mechanisms of these two diseases, ultimately leading to the development of more effective treatment strategies.

Similar to previous studies [53–54], the identification of hub genes and pathways in our study has profound implications for the clinical management of PTC and RA, from diagnostics to therapeutics. The discovery of differentially expressed genes, such as CXCL10 and IL6, presents opportunities for developing diagnostic biomarkers that could indicate disease presence or severity, aiding in early detection and patient stratification. Moreover, these genes have the

potential to act as prognostic indicators, providing insights into disease progression and patient outcomes, with elevated levels of CXCL10 in RA, for example, linked to osteoclast formation and disease severity. The central role of these genes in disease pathogenesis also makes them potential therapeutic targets, where interventions such as small molecule inhibitors or monoclonal antibodies could disrupt inflammatory processes common to both diseases. Our findings further support the shift toward personalized medicine, where treatment plans could be tailored on the basis of the expression levels of these genes, maximizing efficacy and minimizing side effects. The shared pathways between PTC and RA also open new avenues for research, inviting investigations into their mechanistic links and the exploration of these genes in other disease contexts. This understanding could contribute to the development of combination therapies targeting multiple aspects of the disease process, such as the simultaneous modulation of the AMPK and PPAR pathways. Additionally, the identified genes could serve as real-time monitors of treatment response, allowing for dynamic adjustments in patient care. In essence, our identification of shared DEGs not only deepens our comprehension of the molecular foundations of PTC and RA but also lays the groundwork for innovative diagnostic, prognostic, and therapeutic strategies that have the potential to revolutionize their clinical management.

Our study opens several avenues for future research that could significantly enhance our understanding of PTC and RA. The identification of shared DEGs such as CXCL10 and IL6 invites further investigation into their mechanistic roles, building on studies that have underscored the importance of chemokines in immune regulation and cancer progression [55]. Future work could also focus on validating these genes as therapeutic targets, especially considering the impact of pathway targeting observed in recent research [56–58]. The development of personalized medicine approaches, guided by the genetic heterogeneity revealed in studies [59], may lead to more tailored treatments. Additionally, exploring combination therapies that simultaneously modulate multiple pathways, as suggested by research into the synergistic effects of combined treatments [60], could improve patient outcomes. Longitudinal studies that track the expression of these genes may offer insights into disease dynamics and treatment responses, as highlighted by the dynamic nature of gene expression in RA [61]. Finally, the exploration of immune modulation therapies targeting the identified pathways could be a promising direction, inspired by research into the modulation of immune responses by cytokines and chemokines [55]. Collectively, these future research directions, supported by the referenced literature, hold great promise for advancing our knowledge and treatment strategies for PTC and RA.

The novelty of our study emerges from the synergistic application of established bioinformatics methods to explore the shared pathogenesis of PTC and RA. While these methods are common individually, their integrated use provides a unique and comprehensive analysis that has not been previously employed in this context. By combining gene expression profiling, functional enrichment, protein–protein interaction (PPI) network analysis, and transcription factor validation, we identified a core set of DEGs that are pivotal to both diseases. The technical strength of our work lies in the rigorous bioinformatics approach followed by empirical validation using clinical samples, ensuring the reliability and clinical relevance of our findings. Our use of advanced tools such as DESeq2 for data normalization and TRRUST for regulatory network prediction has enabled a robust examination of the molecular interplay between PTC and RA. Moreover, our study introduces a systems biology perspective that surpasses traditional methodologies, offering a holistic view of disease mechanisms and revealing potential therapeutic targets that may have been obscured by more focused analyses. In essence, the innovation of our research is not in the use of common methods but in how we have harnessed them to offer new insights into PTC and RA, setting a new standard for integrative bioinformatics studies in this field.

While our study sheds new light on the shared pathogenesis of PTC and RA, it also underscores the imperative need for sustained research efforts to transcend current methodological constraints and deepen our comprehension of these interconnected diseases. Our pioneering work in verifying common pathogenic genes, such as CXCL10, through retrospective clinical sample analysis lays the groundwork for further exploration of the molecular mechanisms involved in PTC and RA. Nonetheless, we recognize the inherent limitations in our research, including the relatively constrained scope of clinical data for verification, which may influence the robustness of our conclusions, and the necessity for more exhaustive investigations into the functional pathways of genes such as CXCL10. The evolution of computational power and the burgeoning application of machine learning, particularly deep learning, offer promising avenues for overcoming these limitations. By incorporating sophisticated computational models from the literature, including applications of machine learning, deep learning, and other advanced algorithms [62–70], the precision and profundity of bioinformatics analyses can be substantially enhanced. These models can provide a more nuanced understanding of complex biological systems, potentially illuminating intricate PPI network interactions and revealing novel biomarkers that our current datasets could not fully encapsulate. Future studies should leverage these computational advances to analyze larger and more diverse cohorts, leading to more generalized findings and possibly uncovering population-specific biomarkers. The incorporation of multiomics data, including proteomics and epigenetics data, alongside machine learning algorithms could offer a comprehensive molecular landscape, facilitating the development of predictive models for personalized treatment strategies. Moreover, the dynamic progression of diseases requires time-series analysis, which, when combined with machine learning, could offer insights into the temporal changes in disease pathology, guiding the creation of stage-specific therapeutic interventions. In conclusion, our study represents a notable step forward in understanding the shared pathogenesis of PTC and RA. By addressing the limitations of our methodologies and embracing the potential of emerging computational technologies, we are poised to expand our knowledge and bolster our clinical capabilities in managing these intricate diseases.

## Conclusions

In summary, this investigation effectively pinpointed the shared DEGs between PTC and RA via exhaustive enrichment and PPI network analyses. PTC and RA involve several mutual pathogenic processes that are potentially influenced by particular hub genes. The expression of CXCL10 was confirmed in clinical samples, highlighting its crucial function in both PTC and RA. This study offers novel perspectives and sets the stage for additional investigations into the molecular mechanisms involved when PTC is complicated with RA. These discoveries could pave the way for the development of innovative therapeutic approaches aimed at these diseases.

## Supporting information

**S1 Table.  PTA and RA related clinical study summary.**
(DOCX)

## Acknowledgments

None to declare.

## Author contributions

**Conceptualization:** Yingming Liu, Xiangjun Kong, Qianshu Sun, Tianxing Cui, Shengnan Xu, Chao Ding.

**Data curation:** Qianshu Sun, Tianxing Cui, Shengnan Xu.

**Formal analysis:** Yingming Liu, Qianshu Sun, Tianxing Cui.

**Funding acquisition:** Chao Ding.

**Investigation:** Xiangjun Kong, Shengnan Xu, Chao Ding.

**Methodology:** Yingming Liu, Tianxing Cui, Shengnan Xu.

**Project administration:** Yingming Liu, Xiangjun Kong, Shengnan Xu, Chao Ding.

**Software:** Yingming Liu, Xiangjun Kong, Qianshu Sun, Chao Ding.

**Validation:** Qianshu Sun, Tianxing Cui, Chao Ding.

**Visualization:** Xiangjun Kong, Chao Ding.

**Writing – original draft:** Yingming Liu, Xiangjun Kong, Qianshu Sun, Tianxing Cui, Shengnan Xu.

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
