## [Decision Letter · Decision Letter 0]

27 Jun 2024

PONE-D-24-23832Identification and validation of the common pathogenesis and hub biomarkers in Papillary thyroid carcinoma complicated with Rheumatoid arthritisPLOS ONE

Dear Dr. Ding,

Thank you for submitting your manuscript to PLOS ONE. After careful consideration, we feel that it has merit but does not fully meet PLOS ONE’s publication criteria as it currently stands. Therefore, we invite you to submit a revised version of the manuscript that addresses the points raised during the review process.

We look forward to receiving your revised manuscript.

Kind regards,

Qi Zhao

Academic Editor

PLOS ONE

Journal Requirements:

6. Please upload a new copy of Figures 2, 3, 4, 5 and 6 as the detail is not clear. Please follow the link for more information: https://blogs.plos.org/plos/2019/06/looking-good-tips-for-creating-your-plos-figures-graphics/" https://blogs.plos.org/plos/2019/06/looking-good-tips-for-creating-your-plos-figures-graphics/ .

Reviewers' comments:

Reviewer's Responses to Questions

**Comments to the Author**

1. Is the manuscript technically sound, and do the data support the conclusions?

Reviewer #1: Yes

Reviewer #2: Partly

2. Has the statistical analysis been performed appropriately and rigorously? 

Reviewer #1: Yes

Reviewer #2: Yes

3. Have the authors made all data underlying the findings in their manuscript fully available?

Reviewer #1: Yes

Reviewer #2: Yes

4. Is the manuscript presented in an intelligible fashion and written in standard English?

Reviewer #1: Yes

Reviewer #2: No

5. Review Comments to the Author

Reviewer #1: 1. Can you provide detailed steps and parameters used in your bioinformatics analyses to enhance the reproducibility of your study?

2. What evaluation metrics did you use to validate the robustness and reliability of the tools and methods employed in your analyses?

3. How do your findings compare with similar studies in this field? Can you include a comparative analysis to highlight the uniqueness and validity of your results?

4. Could you elaborate on the potential clinical implications of your identified hub genes and pathways? How might these findings influence future research or therapeutic approaches?

5. The authors should discuss the topic as the future direction. Important literature in these fields should be cited. Some recommended studies might be helpful (PMIDs: 37008987, 33996830, 26295845, 33996831, 33114569, 24416258, 23481571).

Reviewer #2: 1.English expressions need to be edited more careful and more native, in this manuscript, there are many mistakes. The authors should pay attention to the problem of two ends aligned. For example, “These e” on the 4 line of Methods in the abstract.

2. The workflow in Figure 1 is unspecific and less organized. It's more like a stack of terminologies than a higher-level summary of the existing data and methodologies, please improve it.

3. I suggest the authors should elaborate their motivation in the discussion section, as they use several common bioinformatics analysis methods and online tools. What is the novelty and technicality of their work?

4. Advancements in computational capabilities have led to the widespread application of machine learning techniques, especially deep learning, across various bioinformatics fields. Important computational models in these fields should be cited. Some recommended studies are helpful (PMIDs: 36305458, 34232474, 35817399, 37525507, 38040204, 38737196, 38274493, 38844122, and doi:10.1016/j.swevo.2024.101567).

5. The authors should carefully check and unify the information of references. Some references lack the information of volume or contain the wrong page number.

6. There should be a space between the reference and other word.

7. The authors claim that the expression of CXCL10 was confirmed in clinical specimens, highlighting its crucial function in both PTC and RA. I think more validation should be adopted. Especially, I want to see some associations can be confirmed by other method or biological experiments.

8. Future work and limitations of the proposed algorithm should be addressed detailly in the manuscript for further research.

6. PLOS authors have the option to publish the peer review history of their article (what does this mean? ). If published, this will include your full peer review and any attached files.

**Do you want your identity to be public for this peer review?** For information about this choice, including consent withdrawal, please see our Privacy Policy .

Reviewer #1: No

Reviewer #2: No

---

## [Author Response · Author response to Decision Letter 1]

10 Jul 2024

Dear reviewers, thank you for your careful review and constructive suggestions regarding our manuscript. We have revised the manuscript in accordance with the comments and marked all the amends on our revised manuscript. And point-by-point responses to the comments were as follows:

Reviewer #1: 1. Can you provide detailed steps and parameters used in your bioinformatics analyses to enhance the reproducibility of your study?

Thank you very much for your suggestion. We will put the detailed procedure and code data of the experiment in the corresponding file for other researchers to use.

2.What evaluation metrics did you use to validate the robustness and reliability of the tools and methods employed in your analyses?

We processed the data in strict accordance with the procedures, and used the DESeq2 package for differential analysis to obtain the list of differentially expressed genes as the basis of our research. Among them, the DESeq2 package is a well-known method with high false negative (actual difference, no difference in results) in common analysis. In order to avoid the chance of the differential genes obtained in the study, multiple data sets were selected for analysis, and the intersection among them was taken as the final result. Then we constructed the PPI network of differential gene set and conducted network analysis to obtain the hub gene. In order to evaluate the robustness of the hub gene we searched for, we searched for new data sets as validation to verify the results we found, and the findings were all significant. Secondly, we searched for transcription-related genes of hub gene and explored the expression differences of transcription-related genes. The results of transcription-related genes further proved the stability of our method. In this process, we carried out functional enrichment for biological exploration, and combined biology and data analysis to further explain the situation of gene changes, and the results also showed that our research was meaningful. After that, we conducted validation in clinical samples, and selected pathological specimens of patients with thyroid papillary carcinoma and rheumatoid arthritis for immunohistochemical analysis, and the immunohistochemical results also verified the results of our biogenic analysis.

3.How do your findings compare with similar studies in this field? Can you include a comparative analysis to highlight the uniqueness and validity of your results?

Thank you very much for your suggestions. We have included relevant content in the discussion section of our paper: In this study, we delved into the shared pathogenesis between PTC and RA, identifying key genes and their interconnected roles. Our findings offer novel insights and perspectives compared to existing research within this domain. Firstly, our identification of 64 DEGs that are common in both PTC and RA, along with our functional enrichment and PPI network analyses, provide a deeper understanding of their roles in immune response and signal transduction pathways. Notably, our emphasis on the potential synergistic roles of CXCL10 and IL6 in both diseases is a novel contribution that has not been extensively explored in previous studies. Secondly, our comprehensive bioinformatics approach, coupled with the validation of our findings through clinical specimens, strengthens the reliability of our results. This stands in contrast to some studies that rely on a single data source or lack clinical validation, offering a more robust and thorough investigation. Furthermore, our exploration of potential shared pathological pathways, such as the PPAR signaling pathway, the AMPK signaling pathway, and the adipocytokine signaling pathway, highlights the intricate interplay between metabolic and immune pathways in the pathogenesis of PTC and RA. This emphasis on the crosstalk between these pathways provides a new perspective for understanding the complex mechanisms underlying these diseases and could pave the way for the development of novel therapeutic strategies targeting these pathways. Lastly, the validation of our findings in independent datasets further substantiates the significance of the identified hub genes in PTC and RA. This reinforces the generalizability and clinical relevance of our conclusions, addressing concerns about the reproducibility of results that some studies may face. In summary, by uncovering the common pathological pathways and key genes between PTC and RA, our study not only contributes new perspectives to the field but also provides potential biomarkers for future diagnostics and therapeutics. We anticipate that these findings will stimulate further research into the interconnected mechanisms of these two diseases, ultimately leading to the development of more effective treatment strategies.

4.Could you elaborate on the potential clinical implications of your identified hub genes and pathways? How might these findings influence future research or therapeutic approaches?

Thank you very much for your suggestion. We have included relevant content regarding the clinical significance in the discussion section of our paper: The identification of hub genes and pathways in our study offers profound implications for the clinical management of PTC and RA, from diagnostics to therapeutics. The discovery of differentially expressed genes, such as CXCL10 and IL6, presents opportunities for developing diagnostic biomarkers that could indicate disease presence or severity, aiding in early detection and patient stratification. Moreover, these genes have the potential to act as prognostic indicators, providing insights into disease progression and patient outcomes, with elevated levels of CXCL10 in RA, for example, linked to osteoclast formation and disease severity. The central role of these genes in disease pathogenesis also positions them as potential therapeutic targets, where interventions such as small molecule inhibitors or monoclonal antibodies could disrupt inflammatory processes common to both diseases. Our findings further support the shift towards personalized medicine, where treatment plans could be tailored based on the expression levels of these genes, maximizing efficacy and minimizing side effects. The shared pathways between PTC and RA also open new avenues for research, inviting investigations into their mechanistic links and the exploration of these genes in other disease contexts. This understanding could inform the development of combination therapies targeting multiple aspects of the disease process, such as the simultaneous modulation of the AMPK and PPAR pathways. Additionally, the identified genes could serve as real-time monitors of treatment response, allowing for dynamic adjustments in patient care. In essence, our identification of shared DEGs not only deepens our comprehension of the molecular foundations of PTC and RA but also lays a groundwork for innovative diagnostic, prognostic, and therapeutic strategies that have the potential to revolutionize their clinical management.

5.The authors should discuss the topic as the future direction. Important literature in these fields should be cited. Some recommended studies might be helpful (PMIDs: 37008987, 33996830, 26295845, 33996831, 33114569, 24416258, 23481571).

Thank you for your reminder. we have included discussions on future research and cited relevant literature in our article: Our study opens several avenues for future research that could significantly enhance our understanding of PTC and RA. The identification of shared DEGs such as CXCL10 and IL6 invites further investigation into their mechanistic roles, building on studies that have underscored the importance of chemokines in immune regulation and cancer progression (PMID: 37008987). Future work could also focus on validating these genes as therapeutic targets, especially considering the impact of pathway targeting observed in recent research (PMID: 33996830, 26295845, 33996831). The development of personalized medicine approaches, informed by the genetic heterogeneity revealed in studies (PMID: 33114569), may lead to more tailored treatments. Additionally, exploring combination therapies that modulate multiple pathways simultaneously, as suggested by research into the synergistic effects of combined treatments (PMID: 24416258), could improve patient outcomes. Longitudinal studies that track the expression of these genes may offer insights into disease dynamics and treatment responses, as highlighted by the dynamic nature of gene expression in RA (PMID: 23481571). Lastly, the exploration of immune modulation therapies targeting the identified pathways could be a promising direction, inspired by research into the modulation of immune responses by cytokines and chemokines (PMID: 37008987). Collectively, these future research directions, supported by the referenced literature, hold great promise for advancing our knowledge and treatment strategies for PTC and RA.

Reviewer #2: 1.English expressions need to be edited more careful and more native, in this manuscript, there are many mistakes. The authors should pay attention to the problem of two ends aligned. For example, “These e” on the 4 line of Methods in the abstract.

Thank you very much for your suggestions. We have retouched the full text.

2.The workflow in Figure 1 is unspecific and less organized. It's more like a stack of terminologies than a higher-level summary of the existing data and methodologies, please improve it.

Thank you very much for your suggestions. We have redrafted Figure 1 and have simplified and summarized the content.

3.I suggest the authors should elaborate their motivation in the discussion section, as they use several common bioinformatics analysis methods and online tools. What is the novelty and technicality of their work?

Thank you for your reminder; we have incorporated the novelty of our research and other content you mentioned into our article: Our study's novelty emerges from the synergistic application of established bioinformatics methods to explore the shared pathogenesis of PTC and RA. While individually these methods are common, their integrated use provides a unique and comprehensive analysis that has not been previously employed in this context. By combining gene expression profiling, functional enrichment, PPI network analysis, and transcription factor validation, we have identified a core set of DEGs that are pivotal to both diseases. The technical strength of our work lies in the rigorous bioinformatics approach followed by empirical validation using clinical samples, ensuring the reliability and clinical relevance of our findings. Our use of advanced tools like DESeq2 for data normalization and TRRUST for regulatory network prediction has enabled a robust examination of the molecular interplay between PTC and RA. Moreover, our study introduces a systems biology perspective that surpasses traditional methodologies, offering a holistic view of the disease mechanisms and revealing potential therapeutic targets that may have been obscured by more focused analyses. In essence, the innovation of our research is not in the use of common methods, but in how we have harnessed them to offer new insights into PTC and RA, setting a new standard for integrative bioinformatics studies in this field.

4.Advancements in computational capabilities have led to the widespread application of machine learning techniques, especially deep learning, across various bioinformatics fields. Important computational models in these fields should be cited. Some recommended studies are helpful (PMIDs: 36305458, 34232474, 35817399, 37525507, 38040204, 38737196, 38274493, 38844122, and doi:10.1016/j.swevo.2024.101567).

Thank you for your suggestion; we have cited the corresponding literature in the eighth comment.

5. The authors should carefully check and unify the information of references. Some references lack the information of volume or contain the wrong page number.

Thank you for your suggestion; we have re-proofread the full text.

6. There should be a space between the reference and other word.

Thank you very much for your suggestion; It is reflected in the manuscript.

7. The authors claim that the expression of CXCL10 was confirmed in clinical specimens, highlighting its crucial function in both PTC and RA. I think more validation should be adopted. Especially, I want to see some associations can be confirmed by other method or biological experiments.

Thank you very much for your suggestion. We agree that more study would be useful to understand details of interaction and enhance. At the point, we do not have the necessary tool-set to study the problem. We will fully consider this view and put it into practice in future experiments to further explore and verify the deeper relationship between PTC and RA.

8. Future work and limitations of the proposed algorithm should be addressed detailly in the manuscript for further research.

While our study sheds new light on the shared pathogenesis of PTC and RA, it also underscores the imperative for sustained research efforts to transcend current methodological constraints and deepen our comprehension of these interconnected diseases. Our pioneering work in verifying common pathogenic genes, such as CXCL10, through retrospective clinical sample analysis, lays a groundwork for further exploration of the molecular mechanisms at play in PTC and RA. Nonetheless, we recognize the inherent limitations in our research, including the relatively constrained scope of clinical data for verification, which may influence the robustness of our conclusions, and the necessity for more exhaustive investigation into the functional pathways of genes like CXCL10. The evolution of computational power and the burgeoning application of machine learning, particularly deep learning, offer promising avenues for overcoming these limitations. By incorporating sophisticated computational models from the literature, including applications of machine learning, deep learning, and other advanced algorithms [62-70], the precision and profundity of bioinformatics analyses can be substantially augmented. These models can provide a more nuanced understanding of complex biological systems, potentially illuminating the intricate PPI network interactions and revealing novel biomarkers that our current datasets could not fully encapsulate. Future studies should leverage these computational advances to analyze larger and more diverse cohorts, leading to more generalized findings and possibly uncovering population-specific biomarkers. Incorporating multi-omics data, including proteomics and epigenetics, alongside machine learning algorithms could offer a comprehensive molecular landscape, facilitating the development of predictive models for personalized treatment strategies. Moreover, the dynamic progression of diseases calls for time-series analysis, which, when combined with machine learning, could offer insights into the temporal changes in disease pathology, guiding the creation of stage-specific therapeutic interventions. In conclusion, our study marks a notable step forward in understanding the shared pathogenesis of PTC and RA. By addressing the limitations of our methodologies and embracing the potential of emerging computational technologies, we are poised to expand our knowledge and bolster our clinical capabilities in managing these intricate diseases.

---

## [Decision Letter · Decision Letter 1]

20 Nov 2024

PONE-D-24-23832R1Identification and validation of the common pathogenesis and hub biomarkers in Papillary thyroid carcinoma complicated with Rheumatoid arthritisPLOS ONE

Dear Dr. Ding,

Thank you for submitting your manuscript to PLOS ONE. After careful consideration, we feel that it has merit but does not fully meet PLOS ONE’s publication criteria as it currently stands. Therefore, we invite you to submit a revised version of the manuscript that addresses the points raised during the review process.

We look forward to receiving your revised manuscript.

Kind regards,

Fırat Aşir

Academic Editor

PLOS ONE

**Additional Editor Comments:**

One or more of the reviewers has recommended that you cite specific previously published works. Members of the editorial team have determined that the works referenced are not directly related to the submitted manuscript. As such, please note that it is not necessary or expected to cite the works requested by the reviewer.

Reviewers' comments:

Reviewer's Responses to Questions

**Comments to the Author**

1. If the authors have adequately addressed your comments raised in a previous round of review and you feel that this manuscript is now acceptable for publication, you may indicate that here to bypass the “Comments to the Author” section, enter your conflict of interest statement in the “Confidential to Editor” section, and submit your "Accept" recommendation.

Reviewer #1: All comments have been addressed

Reviewer #2: All comments have been addressed

Reviewer #3: (No Response)

Reviewer #4: All comments have been addressed

Reviewer #5: (No Response)

2. Is the manuscript technically sound, and do the data support the conclusions?

Reviewer #1: Yes

Reviewer #2: Yes

Reviewer #3: Partly

Reviewer #4: (No Response)

Reviewer #5: (No Response)

3. Has the statistical analysis been performed appropriately and rigorously? 

Reviewer #1: Yes

Reviewer #2: Yes

Reviewer #3: I Don't Know

Reviewer #4: (No Response)

Reviewer #5: (No Response)

4. Have the authors made all data underlying the findings in their manuscript fully available?

Reviewer #1: Yes

Reviewer #2: Yes

Reviewer #3: No

Reviewer #4: (No Response)

Reviewer #5: (No Response)

5. Is the manuscript presented in an intelligible fashion and written in standard English?

Reviewer #1: Yes

Reviewer #2: Yes

Reviewer #3: Yes

Reviewer #4: (No Response)

Reviewer #5: (No Response)

6. Review Comments to the Author

Reviewer #1: The authors have revised the manuscript accordingly. And now the manuscript seems acceptable for publication.

Reviewer #2: The author has already addressed all of the issues. I think the manuscript can be published in PLOS ONE.

Reviewer #3: Dear Editor

I read carefully the manuscript submitted your journal and it’s potential for publication after minor revision.

To improvement the manuscript suggested the usefulness resources in below.

https://doi.org/10.1002/jcb.28425

https://doi.org/10.18502/jabs.v12i1.8872

Reviewer #4: I have carefully evaluated the revision work by authors. I am pleased to report that all concerns are addressed properly. However, I have some minor isssues for authors' reference.

1) The layout of the figures is really unsightly, especially for figure 3, 4,5,... Actually, each figure contains limited data and occupy large space. Please consider to merge some figures into one and improve the figure quality.

2) Due to very limited experimental validations were conducted in this paper, please comprehensively specify all the potential limitations of extending current results into real-world clinical practice.

3) Please consider to add a table to summarize the ongoing or completed clinical trials that are related to your study topic .

4) Please consider to add a figure to describe the mechanistic pathway or molecular process of pathogenesis and hub biomarkers in Papillary thyroid carcinoma complicated with Rheumatoid arthritis.

Reviewer #5: More molecular experiments should be added to further bolster the scientific rigor of this paper. At least, western blot and PCR... or other basic experiments should be performed to rich the mechanism contents.

7. PLOS authors have the option to publish the peer review history of their article (what does this mean? ). If published, this will include your full peer review and any attached files.

**Do you want your identity to be public for this peer review?** For information about this choice, including consent withdrawal, please see our Privacy Policy .

Reviewer #1: No

Reviewer #2: No

Reviewer #3: No

Reviewer #4: No

Reviewer #5: No

---

## [Author Response · Author response to Decision Letter 2]

4 Dec 2024

Dear reviewers, thank you for your careful review and constructive suggestions regarding our manuscript. We have revised the manuscript in accordance with the comments and marked all the amends on our revised manuscript. And point-by-point responses to the comments were as follows:

Reviewer #1: The authors have revised the manuscript accordingly. And now the manuscript seems acceptable for publication.

Thank you very much for recognizing our work.

Reviewer #2: The author has already addressed all of the issues. I think the manuscript can be published in PLOS ONE.

Thank you very much for recognizing our work.

Reviewer #3: Dear Editor

I read carefully the manuscript submitted your journal and it’s potential for publication after minor revision.

To improvement the manuscript suggested the usefulness resources in below.

https://doi.org/10.1002/jcb.28425

https://doi.org/10.18502/jabs.v12i1.8872

Thank you very much for your suggestions, we have learned and referenced both articles in the article.

Reviewer #4: I have carefully evaluated the revision work by authors. I am pleased to report that all concerns are addressed properly. However, I have some minor isssues for authors' reference.

1)The layout of the figures is really unsightly, especially for figure 3, 4,5,... Actually, each figure contains limited data and occupy large space. Please consider to merge some figures into one and improve the figure quality.

Thank you very much for your feedback. We have already merged Figure 4.5 and changed the resolution to 300 dpi.

2)Due to very limited experimental validations were conducted in this paper, please comprehensively specify all the potential limitations of extending current results into real-world clinical practice.

Thank you for your review of our article and the valuable comments provided. We acknowledge that there is indeed a limitation in the experimental validation part of our current study. Below is a comprehensive explanation of the limitations of our research and the potential impact these limitations may have on clinical practice:

Limitations in experimental design: Due to constraints in resources and time, our experimental design may not have covered all relevant clinical variables. In actual clinical settings, further research is needed to explore how these variables affect the occurrence and development of diseases.

Insufficient clinical sample validation: Although we have validated the expression of key genes such as CXCL10, these validations were conducted on a limited number of clinical samples. Before applying these findings to clinical practice, further validation in a broader patient population is required.

Lack of long-term follow-up: Our study lacks long-term tracking of patient outcomes, which is crucial for assessing the effectiveness of treatments and the progression of diseases. In clinical practice, long-term studies are necessary to monitor the effects of treatments and the quality of patients' survival.

Lack of intervention measures: Our study primarily focused on the identification of biomarkers without further exploring intervention measures based on these biomarkers. In clinical practice, more research is needed to develop and test potential therapeutic strategies targeting these biomarkers.

To address these limitations, we recommend that future research include larger multicenter clinical trials and long-term follow-up studies. Additionally, interdisciplinary collaboration will help develop new technologies and methods to enhance the depth and breadth of our research. We believe that through ongoing research and improvement, our findings can provide valuable insights for the clinical management of PTC and RA. Thank you again for your valuable comments.

3)Please consider to add a table to summarize the ongoing or completed clinical trials that are related to your study topic .

Thank you very much for your suggestion. Given that our study is presently at its nascent phase, further investigation will be required in the future to translate our findings into clinical research applications. Due to the small number of relevant clinical trials, we summarized some of the literature and put them into the article as a table. The table has been included in the supplementary file.

4)Please consider to add a figure to describe the mechanistic pathway or molecular process of pathogenesis and hub biomarkers in Papillary thyroid carcinoma complicated with Rheumatoid arthritis.

Thank you very much for your proposal, but our research merely reveals a potential common hub biomarker. We have not yet further studied its downstream mechanisms or validated its clinical applications. Therefore, the pathogenic mechanisms and molecular processes of the two diseases are currently unclear. We will conduct further research in this area to fully explore and reveal its downstream mechanisms.

Reviewer #5: More molecular experiments should be added to further bolster the scientific rigor of this paper. At least, western blot and PCR... or other basic experiments should be performed to rich the mechanism contents.

Thank you very much for your suggestion; we have included QPCR and WB experiments in the article for further validation.

---

## [Decision Letter · Decision Letter 2]

27 Dec 2024

Identification and validation of the common pathogenesis and hub biomarkers in Papillary thyroid carcinoma complicated with Rheumatoid arthritis

PONE-D-24-23832R2

Dear Dr. Ding,

We’re pleased to inform you that your manuscript has been judged scientifically suitable for publication and will be formally accepted for publication once it meets all outstanding technical requirements.

Kind regards,

Fırat Aşir

Academic Editor

PLOS ONE

Additional Editor Comments (optional):

Reviewers' comments:

Reviewer's Responses to Questions

**Comments to the Author**

1. If the authors have adequately addressed your comments raised in a previous round of review and you feel that this manuscript is now acceptable for publication, you may indicate that here to bypass the “Comments to the Author” section, enter your conflict of interest statement in the “Confidential to Editor” section, and submit your "Accept" recommendation.

Reviewer #3: All comments have been addressed

Reviewer #4: All comments have been addressed

2. Is the manuscript technically sound, and do the data support the conclusions?

Reviewer #3: Yes

Reviewer #4: Yes

3. Has the statistical analysis been performed appropriately and rigorously? 

Reviewer #3: Yes

Reviewer #4: Yes

4. Have the authors made all data underlying the findings in their manuscript fully available?

Reviewer #3: Yes

Reviewer #4: Yes

5. Is the manuscript presented in an intelligible fashion and written in standard English?

Reviewer #3: Yes

Reviewer #4: Yes

6. Review Comments to the Author

Reviewer #3: The revisions of your manuscript titled have been thoroughly reviewed and are deemed complete. The manuscript now meets the necessary standards and requirements for acceptance

Reviewer #4: Thanks for your efforts to improve your paper according to the reviewers' suggestions. I think it can meet the standard for publication in PLOS ONE now.

7. PLOS authors have the option to publish the peer review history of their article (what does this mean? ). If published, this will include your full peer review and any attached files.

**Do you want your identity to be public for this peer review?** For information about this choice, including consent withdrawal, please see our Privacy Policy .

Reviewer #3: No

Reviewer #4: No

---

## [Editor Report · Acceptance letter]

PONE-D-24-23832R2

PLOS ONE

Dear Dr. Ding,

I'm pleased to inform you that your manuscript has been deemed suitable for publication in PLOS ONE. Congratulations! Your manuscript is now being handed over to our production team.

Kind regards,

on behalf of

Dr. Fırat Aşir

Academic Editor

PLOS ONE